# Optimizing cognitive and behavioral approaches for perinatal depression: A systematic review and meta-regression analysis

Ahmed Waqas[1] , Syeda Wajeeha Zafar[2], Parveen Akhtar[3], Sadiq Naveed[4] and Atif Rahman[1]

[1]Department of Primary Care & Mental Health, Institute of Population Health, University of Liverpool, Liverpool, UK; [2]Global Institute of Human Development, Shifa Tameer-e-Millat University, Islamabad, Pakistan; [3]Department of Psychology, Capital University of Science and Technology, Islamabad, Pakistan and [4]Department of Psychiatry, Eastern Connecticut Health Network, Manchester, CT, USA

## Overview Review

**Keywords:**
cognitive behavioral therapy; CBT; perinatal depression

**Author for correspondence:**
Ahmed Waqas,
Email: ahmed.waqas@liverpool.ac.uk

## Abstract

Cognitive behavioral therapies (CBT) have been demonstrated efficacious in treating perinatal depression (PND). This has been demonstrated in several meta-analyses of randomized controlled trials and quasi-experimental studies. However, there is a need for up-to-date meta-analytical evidence providing reliable estimates for CBT's effectiveness in treating and preventing PND. Furthermore, with the world moving toward precision medicine, approaches require a critical synthesis of psychotherapies, especially to unpack their mechanisms of action and to understand what approaches work best for whom. Therefore, the present systematic review and meta-regression analyses aim to answer these research questions.

We searched six academic databases through February 2022 and identified 56 studies for an in-depth review. Using pretested data extraction sheets, we extracted patient-level and intervention-level characteristics and effect size data from each study. Random-effects meta-analyses and mixed-effect subgroup analyses were run to delineate the effectiveness and moderators of CBT interventions for PND, respectively. CBT-based interventions yielded a strong effect size (SMD = −0.74, 95% confidence interval [CI]: −0.91 to −0.56, $n$ = 9,722) in alleviating depressive symptoms. These interventions were effective across different delivery formats (individual, group, and electronic) and could be delivered effectively by specialists and nonspecialists. Longer duration CBT interventions may not necessarily be more effective than shorter ones. Moreover, CBT-based interventions should consider including various behavioral ingredients to maximize intervention benefits.

## Impact statement

Perinatal depression is very prevalent worldwide. It is associated with poor maternal and infant health outcomes and thus, a significant public health concern. Cognitive behavioral (CB) therapy is an evidence-based and one of the most effective treatments for perinatal depression. This systematic review and meta-analysis provide an overview of interventional research testing different CB approaches for perinatal depression. It synthesizes findings about the development of CB-based approaches delivered either individually, in groups or electronically. Thereafter, using established frameworks, this review also dissects the interventions into their components. Quantitative evidence is provided regarding the factors which could improve or worsen the efficacy of these interventions. These include but are not limited to the characteristics of women undergoing CB treatment, the format of delivery, and approaches utilized in these intervention programs. It is hoped that this synthesis of literature would guide researchers, clinicians, and implementors in better delivery of CB approaches for perinatal depression in different settings.

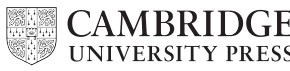

## Background

Perinatal depression (PND) is a public health priority due to its high prevalence and ill effects on child health (Husain et al., 2006; Gelaye et al., 2016; Anderson et al., 2017; Bowers et al., 2021). It is one of the most common mental disorders among perinatal women and is studied widely in low- and middle-income countries (LMICs) (Gelaye et al., 2016). In LMICs, approximately 25.3% of antenatal women and 19% of postpartum women report depressive symptoms (Gelaye et al., 2016). Women with PND are at a higher risk of developing perinatal complications, including intrauterine growth retardation, preterm deliveries, low birth weight, and infectious illnesses

among their infants (Gelaye et al., 2016). In addition, untreated PND affects child health postnatally, leading to poorer growth, neurodevelopmental, socioemotional, and academic outcomes (Dubowitz et al., 2002; Ashman et al., 2008; Betts et al., 2015; Bao et al., 2016; Fanti and Kimonis, 2017; Netsi et al., 2018; Chae et al., 2020; Bowers et al., 2021). Therefore, strategies to address maternal mental health are increasingly becoming the focus of maternal and child public health initiatives, especially in LMICs (Rahman et al., 2008; Rahman et al., 2018; Sikander et al., 2019a; Sikander et al., 2019b).

Fortunately, efficacious preventive and treatment interventions exist for PND in the form of psychological and psychosocial therapies (Sockol, 2015; Rahman et al., 2018; Li et al., 2022; Waqas et al., 2022b). Experimental evidence, however, is still lacking for pharmacotherapies for PND (Howard and Khalifeh, 2020; Brown et al., 2021). Several meta-analyses of randomized controlled trials (RCTs) have repeatedly shown that cognitive-behavioral therapies (CBTs) are among the most efficacious treatments for PND (Sockol, 2015; Rahman et al., 2018; Li et al., 2022; Waqas et al., 2022b). For instance, strong effect sizes were reported for CBT-based treatment interventions for PND Standardized Mean Difference (SMD = 0.65, 95% confidence interval [CI]: 0.54–0.76) (Sockol, 2015). However, CBT interventions have yielded weak-to-moderate strength effect sizes in the prevention of PND (SMD = 0.39, 95%CI: 0.17–0.60) (Sockol, 2015). These therapies are also acceptable among the stakeholders and end-consumers in LMIC and, thus, suitable for large-scale implementation (Rahman et al., 2018).

Research evidence demonstrates CBT interventions' adequate effectiveness, utility, and implementation (Sockol, 2015; Rahman et al., 2018; Li et al., 2022; Waqas et al., 2022b). However, there is a paucity of evidence delineating what works and for whom. Answering these questions is important to optimize psychotherapies for different populations, for example, by choosing the right treatments for the right candidates (Delgadillo et al., 2022). An increasing body of research has shown that these treatments work in different settings (Sockol, 2015). However, there is little research evidence on how and for whom these interventions work (Cuijpers et al., 2019; Furukawa et al., 2021). Thus, delineating the mechanistic pathways of different psychotherapeutic treatments has gained priority in the research agenda for depression (Huibers et al., 2020). Mediation research is an important tool to understand how psychotherapies work, while prediction and moderation research help identify for whom these interventions work (Huibers et al., 2020). Using these tools, we can attempt to unpack the black box of psychological therapies, a challenge for the field identified as early as 1967 by Paul (1967) and Huibers et al. (2020).

More recently, two complementary research streams in psychotherapy have emerged: one that focuses on harmonizing terminology across different schools of psychotherapies (Chorpita et al., 2005; Chowdhary et al., 2014; Singla et al., 2017), and the other focuses on empirical causal processes of change brought about by psychological interventions (Singla et al., 2021). Important research in the former domain includes the works of Chorpita et al. (2005) and Abraham and Michie (2008), who sought to harmonize the taxonomy of treatment strategies utilized across different psychotherapies. Based on this work, researchers have posited that there are commonalities between different forms of psychotherapies. While having different theoretical underpinnings, these psychotherapies may work through similar mechanisms.

Two classes of therapeutic ingredients of psychotherapies have been posited: specific and nonspecific or common ingredients (Singla et al., 2017). Specific active ingredients emerge from the theoretical models of different psychotherapies. For instance, cognitive behavior therapy is based on cognitive theory and is hypothesized to work through challenging and changing maladaptive thought patterns or cognitive schemas, while behavioral therapies work by correcting maladaptive behaviors. Similarly, interpersonal psychotherapy is hypothesized to act through interpersonal change mechanisms (Chorpita et al., 2005; Kahl et al., 2012; Cuijpers et al., 2019; Huibers et al., 2020). However, common ingredients or elements include techniques used by therapists during the delivery of therapy sessions, e.g., building rapport and empathy or helping the client to identify sources of social support. These common active ingredients are shared across all forms of psychotherapy. Rosenzweigh cites these common ingredients as the primary reason for comparable effect sizes across different psychotherapies (Rosenzweigh, 1936; Eyesenck, 1955).

Even after decades of research, none of the theories has yielded conclusive empirical evidence, and the black box of psychotherapies remains unpacked. Moreover, there is also a paucity of evidence on optimizing and personalizing treatment with psychotherapies. Therefore, the present systematic review and meta-regression analysis aims to:

i.   Assess the effectiveness of CBT-based interventions for the prevention and treatment of PND.
ii.  Explore the settings in which these interventions work the best.
iii. Explore the individual level and intervention level factors driving PND's prognosis among women undergoing CBT.
iv.  Explore the active ingredients of CBT interventions for PND.

## Methods

### Search strategy

This systematic review and meta-analysis have been conducted per the Preferred Reporting Items for Systematic reviews and Meta-Analyses (PRISMA) guidelines (Page et al., 2021). Before the conduct of this review, its protocol was registered on the PROSPERO database (Waqas and Rahman, 2022). The current systematic review does not report findings concerning secondary outcomes mentioned in the PROSPERO protocol. Using a pretested search strategy (Supplementary Table 1), we searched six academic databases, including PubMed, Medline, Web of Science, PsycInfo, Cochrane central registry of trials, and CINAHL, through February 2022.

### Inclusion & exclusion criteria

We included all randomized and cluster RCTs that reported the effectiveness of cognitive, behavioral, and third-wave psychotherapeutic interventions as standalone or as part of complex multi-component interventions (Supplementary Table 2). We included CBT-based interventions for PND, delivered during the antenatal period and up to 1-year postnatal. Those trials were considered that reported either the rate of PND or symptom severity of perinatal depressive symptoms as a primary outcome. Preventive interventions were considered for both indicated (populations with prodromal symptoms) and targeted (at-risk) populations. However for treatment interventions, we included those which recruited perinatal women who were either screened positive for PND using psychometric scales or diagnosed clinically using

International Statistical Classification of Diseases and Related Health Problems (ICD) or The Diagnostic and Statistical Manual of Mental Disorders (DSM) clinical diagnostic criteria. Interventions conducted among peripartum women with medical comorbidities were also considered. When available, we also reviewed intervention manuals and secondary publications associated with the eligible RCTs. This was done to aid in synthesizing evidence on the active ingredients of CBT interventions.

We excluded studies that did not report PND (rates of diagnoses or severity of symptoms) as an outcome. We also excluded studies not available in the English language and short formats of publications such as brief reports, letters to editors, conference papers, and abstracts.

### Outcomes

As primary outcomes, we considered on either rate of PND assessed using clinical criteria of diagnoses or scores on valid and reliable psychometric scales; assessed post-intervention. This review does not report findings pertaining to secondary outcomes outlined in the PROSPERO protocol.

### Study selection procedures and data extraction

Teams comprising two independent reviewers screened database records against inclusion and exclusion criteria using a two-phased approach (titles and abstracts followed by full texts). After the identification of studies fulfilling the eligibility criteria, data on characteristics of intervention and study samples were extracted. Study-level characteristics included the year of publication, study design, type of control group, and inclusion and exclusion criteria. However, patient-level characteristics included mean age, the proportion of participants belonging to minority ethnic groups and lower income class, parity, family structure, and intervention timing (antenatal or gestational age if available, or postpartum period). We also cataloged intervention-level characteristics such as the scope of the intervention (targeted prevention, indicated prevention, and treatment), the theoretical underpinning of interventions, the format of delivery (individual, group, electronic), setting of intervention, delivery agent (specialist and nonspecialist), and the number of sessions of intervention. These variables were selected a priori as described in the systematic review protocol (Waqas and Rahman, 2022).

### Taxonomy of interventions: Distillation & matching framework

This exercise was done to delineate different elements and active ingredients of cognitive behavioral interventions included in this review. It is based on the premise that interventions to improve mental health are varied and may comprise: (i) a combination of specific or nonspecific active ingredients underpinned by a single theory-based approach, often called a therapy (e.g., CBT) or (ii) a combination of elements drawn from different theories, forming a multicomponent intervention or eclectic therapy. An additional complication is that multicomponent interventions usually comprise ingredients that may be derived from another discipline, e.g., CBT may be delivered in tandem exercise or yoga. All this creates a problem for the field as it is important for policymakers to know which interventions provide the best evidence for effectiveness and feasibility (Chorpita et al., 2005; Abraham and Michie, 2008; Michie et al., 2013; Cuijpers et al., 2019; Huibers et al., 2020; Furukawa et al., 2021). Furthermore, it also complicates the understanding of

mediational or causal mechanisms that drive an intervention's efficacy.

To decompose the CBT-based interventions into their components or active ingredients, we utilized the distillation and matching framework for psychotherapies devised by Chorpita et al. (2005). This approach was further informed by Michie and colleagues' hierarchically clustered taxonomy of behavior change techniques (Abraham and Michie, 2008; Michie et al., 2013). These frameworks were used to harmonize the definitions of active ingredients across the studies included in this review. To devise a hierarchal taxonomy suitable for this review, we used the definitions proposed by the Institute of Medicine's framework for psychotherapies (England et al., 2015). The hierarchy comprised three levels: elements, strategies, and active ingredients. We defined the elements as either specific or nonspecific. Nonspecific elements are fundamental engagement strategies (e.g., showing empathy) and are essential for building an effective client-therapist alliance. Specific elements are unique to a particular theoretical orientation underpinned by behavioral, cognitive, interpersonal, and emotional domains. This categorization is recommended by Singla et al. and widely adopted by the stakeholders (Rahman et al., 2018; Waqas et al., 2022b). All these elements and active ingredients have been defined in the World Health Organization's guidelines for preventing and treating PND and anxiety (Rahman et al., 2018) and presented here for review. The finalized hierarchy of active ingredients comprised 58 most utilized behavior change techniques and treatment elements (Supplementary Tables 3 and 4). Using the above frameworks, we could also harmonize and standardize strategies utilized across different disciplines and theories. For example, "thought records" in CBT were considered similar to "mood ratings" in interpersonal psychotherapy (England et al., 2015).

This phase was conducted by three experts trained in clinical psychology and psychiatry at postgraduate levels. The reviewers evaluated the content of the interventions as detailed in the trial papers and associated manuals (if available) to identify commonly utilized approaches.

### Risk of bias

The risk of bias among RCTs was assessed using the Cochrane tool for risk of bias assessments (Higgins et al., 2019). It was assessed across five domains, including the method for random sequence generation, allocation concealment, blinding of outcome assessment, attrition bias, and selective reporting. We did not rate risk across blinding of participants and personnel domain as it is challenging to maintain during trials of psychotherapies.

### Data analysis

We conducted a meta-analysis for depressive symptoms according to psychometric scales and the rate of perinatal depressive disorders (ICD/DSM criteria) assessed after the intervention. Findings on secondary outcomes were only synthesized narratively. For continuous outcomes about depressive symptoms severity on psychometric scales, we extracted the mean (standard deviation [SD]) and sample size of intervention and control groups. For binary outcomes, we extracted both groups' number of events and sample sizes. In case scores on psychometric scales were presented as binary outcomes in studies, we converted them to standardized mean differences using the following formula: $\text{SMD} = \sqrt{3}/\pi \ln \text{OR}$ (Higgins et al., 2019).

We expected a high clinical heterogeneity in the eligible studies due to varied approaches for the assessment of clinical outcomes, theoretical underpinnings of included therapies, and population studies. Therefore, we utilized random effects (Der Simonian & Laird method) to pool data across the studies. Study-level and pooled effect sizes were visualized as a forest plot. Sensitivity analyses were conducted to adjust meta-analytical estimates for outliers. Publication bias in the study was assessed statistically using Egger's regression and visualized as Begg's funnel plot (Thornton and Lee, 2000). To identify moderators of effect sizes, we conducted subgroup analyses for study, intervention, and patient-level variables if reported in more than four studies (Borenstein et al., 2021). Meta-regression was done to assess the association of quantitative variables with effect size. To ensure optimum power, meta-regression was only performed when continuous variables were reported in at least 10 studies. (Borenstein et al., 2021).

## Results

### Screening process

The electronic database searches yielded 515 titles and abstracts, out of which 116 duplicate records were removed using Endnote. Out of 399 titles and abstracts, 323 records were excluded after assessing their titles and abstracts against the eligibility criteria for this review. Finally, full texts of 76 studies were appraised, out of which 34 were excluded. A total of 42 studies were eligible to be included in the review. The main reasons for exclusion were non-RCT/cRCT study design ($n = 30$), intervention not for PND ($n = 2$), and short forms of publication ($n = 2$). Fourteen studies were included after the manual screening of bibliographies of included studies and consultations with experts (Figure 1).

Among these 56 studies, there were 59 interventions. Among the included studies, a high proportion of the interventions were

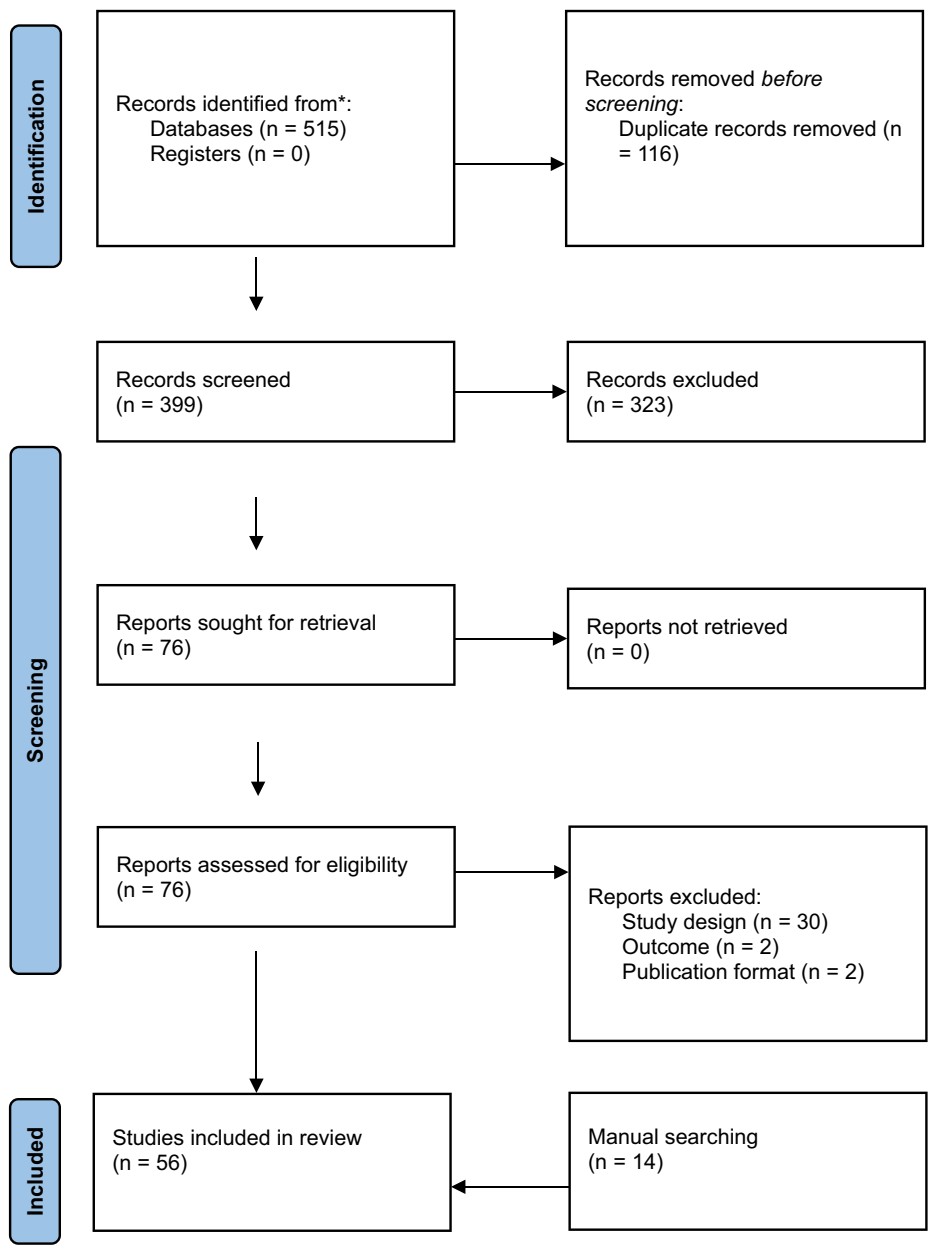

**Figure 1.** PRISMA flowchart showing the process of inclusion of studies.

delivered individually ($n = 24$), followed by group ($n = 25$) and electronic ($n = 10$) delivery format. These interventions were tested among participants with a mean age of 28.48 years (2.99), married ($\bar{x} = 66.8\%$, SD = 31.95), and ($\bar{x} = 48.26\%$, SD = 18.03). Among the participants in included studies, approximately 41% reported low-income levels (SD = 22.32) and poor education ($\bar{x} = 35.25$, SD = 23.69).

### Quality of trials

These interventions were tested in generally high-quality trials, where the random sequence generated was rated at low risk of bias among 41 studies, allocation concealment ($n = 29$), blinding of outcomes assessment ($n = 26$), attrition bias ($n = 35$), and selective reporting ($n = 56$). The risk of bias was unclear for allocation concealment in 27 studies, blinding of outcome assessment ($n = 27$), attrition bias ($n = 13$), and random sequence generation ($n = 10$) (Figure 2).

### Interventions delivered to individuals

Among these interventions (Supplementary Table 5), nine were delivered during the antenatal period (Cho et al., 2008; Silverstein et al., 2011; Hayden et al., 2012; Ammerman et al., 2013; Burns et al., 2013; Dimidjian et al., 2014, 2016; Yazdanimehr et al., 2016; Nejad et al., 2021), followed by postnatal ($n = 8$) (Chabrol et al., 2002; Cooper et al., 2003; Morrell et al., 2009; Hou et al., 2014; Ngai et al., 2015; Kordi et al., 2018; Van Horne et al., 2021) and both periods ($n = 7$) (Prendergast and Austin, 2001; McKee et al., 2006; Rahman et al., 2008; O'Mahen et al., 2013a; Trevillion, 2014; Rahman et al., 2018; Tandon et al., 2018; Sikander et al., 2019b). Eleven interventions were delivered in communities, especially through home visits (Prendergast and Austin, 2001; Chabrol et al., 2002; Cooper et al., 2003; Rahman et al., 2008; Morrell et al., 2009; Ammerman et al., 2013; Burns et al., 2013; Tandon et al., 2018; Sikander et al., 2019a; Van Horne et al., 2021), three in multiple settings (McKee et al., 2006; Silverstein et al., 2011; Dimidjian et al., 2017), while the rest were delivered in healthcare settings (clinic or hospital) (Cho et al., 2008; Hayden et al., 2012; O'Mahen et al., 2013a; Hou et al., 2014; Trevillion, 2014; Ngai et al., 2015; Dimidjian et al., 2016; Yazdanimehr et al., 2016; Kordi et al., 2018; Nejad et al., 2021). A higher proportion of studies utilized The Edinburgh Postnatal Depression Scale (EPDS) for outcome assessment ($n = 12$), followed by The Beck Depression Inventory (BDI) ($n = 6$), The Hamilton Rating Scale for Depression (HDRS) ($n = 2$), The Patient Health Questionnaire (PHQ-9) ($n = 2$), The Quick Inventory of Depressive Symptomatology (QIDS), and The Depression Anxiety Stress Scales (DASS-21) ($n = 2$).

A total of 18 interventions were tested for PND treatment and underpinned by CBT ($n = 16$). Three trials tested Problem solving therapy (PST) (Silverstein et al., 2011; Kordi et al., 2018; Van Horne et al., 2021), mindfulness-based stress reduction or cognitive therapy ($n = 3$; Dimidjian et al., 2016; Yazdanimehr et al., 2016; Nejad et al., 2021), and Behavioral activation (BA) therapy ($n = 1$) (Dimidjian et al., 2017). These interventions were delivered by either specialists ($n = 14$), nonspecialists ($n = 8$), or multidisciplinary teams ($n = 2$). Delivery agents reported diverse disciplinary backgrounds and experience in the delivery of care. Nonspecialists ranged from peers (Sikander et al., 2019a), health visitors (Morrell et al., 2009; Tandon et al., 2018), and allied health professionals such as lady health workers (Rahman et al., 2008), midwives (Ngai et al., 2015), early childhood nurses (Prendergast and Austin, 2001), and social workers (McKee et al., 2006; Hayden et al., 2012; Ammerman et al., 2013; Van Horne et al., 2021) and graduate students in social work, public health, and medical sciences (Silverstein et al., 2011). While delivery agents specializing in mental health included practising clinical psychologists, graduate students, recent graduates (Chabrol et al., 2002; Cho et al., 2008; Burns et al., 2013), counselors, and well-being practitioners (Hou et al., 2014; Trevillion, 2014). Half of these interventions ($n = 12$) were integrated into healthcare settings (Ammerman et al., 2013; Dimidjian et al., 2016; Dimidjian et al., 2017; Hou et al., 2014; Morrell et al., 2009; Ngai et al., 2015; Prendergast J, 2001; Rahman et al., 2008; Sikander et al., 2019a; Tandon et al., 2018; Trevillion, 2014; Van Horne et al., 2021).

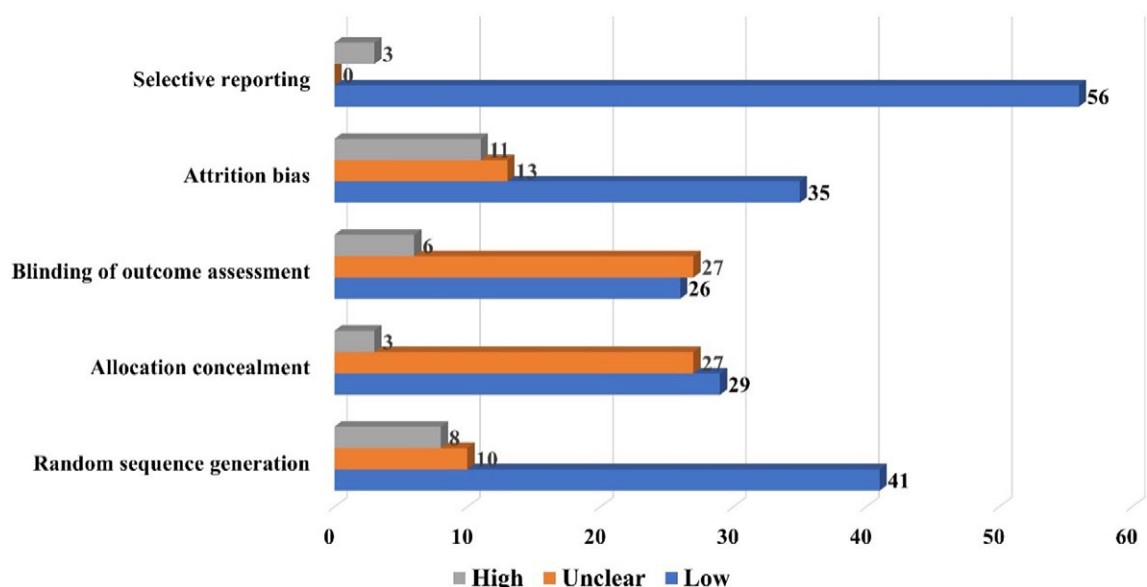

**Figure 2.** Risk of bias in included studies.

The number of sessions ranged from one for prevention intervention by Chabrol et al., (2002) to 16 for treatment (Thinking Healthy Programme) of PND (Rahman et al., 2008). Among nonspecific interventions, the most frequently utilized nonspecific active ingredients were active listening ($n = 10$), empathy ($n = 9$), collaboration ($n = 9$), and inciting social support ($n = 9$) and normalization ($n = 7$). Assigning homework ($n = 8$) and goal setting ($n = 8$) were the most frequently utilized in-session techniques.

Among specific ingredients, interpersonal strategies were frequently utilized, including identifying and eliciting social support ($n = 14$), communication skills ($n = 11$), and identifying affect ($n = 10$). Among behavioral strategies, problem-solving ($n = 16$), relaxation ($n = 7$), emotional regulation and stress management, and decision-making ($n = 5$ each) were frequently utilized. Essential cognitive strategies were identifying thoughts and behaviors and their links ($n = 19$), cognitive restructuring ($n = 16$), self-awareness ($n = 8$), and mood monitoring ($n = 7$). Caregiver coping ($n = 8$), parent–child interaction ($n = 6$), and psychoeducation regarding birth procedures or specific health areas of children ($n = 6$) were also important (Supplementary Figures 1 to 3).

### Intervention delivered in groups

Among these 25 interventions (Supplementary Table 6), 14 were delivered antenatally (Brugha et al., 2000; Austin et al., 2008; Futterman et al., 2010; Lara et al., 2010; Le et al., 2011; Kozinszky et al., 2012; Kaaya et al., 2013; Leung et al., 2013; Bittner et al., 2014; Jesse et al., 2015; Van Ravesteyn et al., 2018; Khamseh et al., 2019; Zemestani and Fazeli Nikoo, 2019), postnatally ($n = 7$; Hagan et al., 2004; Milgrom et al., 2005; Graciela Rojas et al., 2007; Christine Puckering et al., 2010; Mao et al., 2012; Leung et al., 2016; Van Lieshout et al., 2022), and four during both periods (Muñoz et al., 2007; Tandon et al., 2014; Ngai et al., 2019). Only three of these interventions were conducted in communities (Muñoz et al., 2007; Tandon et al., 2014; Van Lieshout et al., 2022), while the rest were conducted in healthcare settings. EPDS was the most frequently utilized scale for outcome assessment, followed by BDI I/II ($n = 5$). Seven interventions were delivered by specialists, 14 by nonspecialists, and four by multidisciplinary teams. Delivery agents were heterogeneous in terms of disciplines and experience and included counseling or clinical psychologists, academics and doctoral students in psychology, nurses, midwives, doctors, obstetricians, social workers, occupational therapists, art therapists, infant mental health specialists, and peers. Thirteen of these interventions were integrated into healthcare systems, while the rest were delivered as standalone.

A total of 13 interventions were tested for treatment and 12 for prevention of PND. Twenty trials tested classical CBT interventions, PST ($n = 2$), psychoeducation ($n = 2$), and Mindfulness-based cognitive therapy (MCBT) ($n = 1$). The sessions ranged from 1 (Ngai et al., 2019) to 14 (Christine Puckering et al., 2010). Among group therapies, the most frequently utilized nonspecific ingredients were inciting social support ($n = 12$), normalization ($n = 9$), and involvement of significant other ($n = 6$). Most frequently employed in-session techniques were assigning homework ($n = 11$), goal setting ($n = 13$), and interpersonal focus ($n = 9$). Among interpersonal strategies, the most frequently utilized ingredients were identifying affect ($n = 15$), identifying and eliciting social support ($n = 13$), and communication skills ($n = 11$). Problem-solving ($n = 18$), relaxation ($n = 16$), and stress management ($n = 13$) were most frequently utilized behavioral

ingredients. Identifying thoughts, behaviors, and their links ($n = 18$), cognitive restructuring ($n = 13$), and mood monitoring ($n = 6$) were important cognitive ingredients. Caregiver coping skills ($n = 8$) and parent–child interaction coaching ($n = 6$) were imparted in a small proportion of trials (Supplementary Figures 4 to 6).

### Interventions delivered online

Seven of these interventions (Supplementary Table 7) were tested during postpartum (O'Mahen et al., 2013b; Milgrom et al., 2016; Wozney et al., 2017; Loughnan et al., 2019a; Fonseca et al., 2020; Jannati et al., 2020; Van Lieshout et al., 2021) and three during the antenatal period (Forsell et al., 2017; Duffecy et al., 2019; Loughnan et al., 2019b). Two of these interventions were for the prevention of PND (Duffecy et al., 2019; Fonseca et al., 2020), while the rest were treatment interventions. All interventions were designed to be used by individuals, except Duffecy et al. and Van Lieshout et al. who delivered to groups of participants (Duffecy et al., 2019; Van Lieshout et al., 2021). Only two of these interventions were guided either by specialist mental health professionals (Van Lieshout et al., 2021) or nonspecialists (Wozney et al., 2017). The number of sessions of interventions ranged from 1 (Van Lieshout et al., 2021) to 16 (Duffecy et al., 2019). Only one intervention was integrated into healthcare settings (Forsell et al., 2017).

All interventions were based on CBT except Fonseca et al. and O'Mahen et al., who tested Acceptance and Commitment Therapy (CBT-ACT) and BA-based interventions (O'Mahen et al., 2013b; Fonseca et al., 2020). Among these interventions, inciting social support ($n = 6$) was the most frequently utilized nonspecific ingredient. Identifying affect ($n = 9$), identifying and eliciting social support ($n = 8$), and communication skills training ($n = 7$) were important interpersonal strategies. Problem-solving ($n = 7$), relaxation ($n = 7$), and self-monitoring were important behavioral approaches. Among cognitive approaches, identifying thoughts ($n = 10$), cognitive restructuring ($n = 8$), and mood monitoring ($n = 7$) were important elements. Assigning homework ($n = 6$) was frequently employed in-session technique. Caregiver coping and parent–child interaction coaching were utilized in four interventions.

None of the interventions employed reinforcement-oriented active ingredients. Four interventions included information on caregiver coping skills (O'Mahen et al., 2013b; Milgrom et al., 2016; Duffecy et al., 2019; Fonseca et al., 2020) and four included parent–child interaction coaching (O'Mahen et al., 2013b; Milgrom et al., 2016; Wozney et al., 2017; Duffecy et al., 2019). Psychoeducation, either on birth procedures, nutrition, breastfeeding, or sexual behaviors was provided in only two interventions (Milgrom et al., 2016; Duffecy et al., 2019). Nutrition and substance use-related counseling were each provided in one study (Wozney et al., 2017; Duffecy et al., 2019). Supplementary Tables 4−6 present the characteristics of studies included in this section. Supplementary Figures 7−9 present active ingredients utilized in online interventions.

### Meta-analysis: Effectiveness

CBT-based interventions yielded a strong effect size (SMD = −0.74, 95% CI: −0.91 to −0.56, $n = 9{,}722$) in alleviating depressive symptoms. There was evidence of substantial heterogeneity in effect sizes across studies ($I^2 = 92.65\%$, $p < 0.001$, and $Q = 775.03$). Sensitivity analysis did not reveal any substantial changes in effect size after removing outliers. There was substantial publication bias,

as evidenced by the funnel plot (Supplementary Figure 10) and Egger's regression statistic ($p = 0.009$). Duval and Tweedie's trim and fill method adjusted pooled effect size for publication bias. After trimming 13 studies to the left of the mean, the adjusted SMD was $-0.95$ (95% CI: $-1.14$ to $-0.76$).

Forest plots were developed separately according to the mode of delivery of interventions. Interventions delivered electronically ($n = 9$) yielded strong effect sizes (SMD $= -1.12$, 95% CI: $-1.80$ to $-0.63$, $n = 1,218$) (Figure 3). There was substantial evidence of heterogeneity across studies ($I^2 = 93.99\%$, $p < 0.001$, and $Q = 133.12$). There was no evidence of publication bias (Egger's regression $p = 0.65$). Intervention delivered to individuals (Figure 4) also yielded strong effect sizes in favor of the intervention group (SMD $= -0.63$, 95% CI: $-0.81$ to $-0.44$, $n = 3,589$).

There was evidence of significant heterogeneity across the studies ($I^2 = 80.99\%$, $p < 0.001$, and $Q = 121.02$). There was evidence for significant publication bias (Egger's regression $P = 0.06$), which after adjustment led to a higher effect size (SMD $= -0.75$, 95% CI: $-0.94$ to $-0.55$). Interventions delivered among groups (Figure 5) also yielded strong effect sizes in favor of the intervention group (SMD $= -0.67$, 95% CI: $-0.96$ to $-0.38$, $n = 4,915$). Statistical heterogeneity was substantial ($I^2 = 94.59\%$, $p < 0.001$, and $Q = 443.90$). There was some evidence of publication bias (Egger's regression $p = 0.10$), with the trim and fill method yielding a higher adjusted effect size (SMD $= -1.00$, 95% CI: $-1.33$ to $-0.67$). Sensitivity analysis did not reveal any substantial changes in effect size after removing outliers in any of the above analyses.

| Study name | Subgroup within study | Outcome | Std diff in means | Lower limit | Upper limit | Intervention | Control | Std diff in means and 95% CI | Relative weight |
|---|---|---|---|---|---|---|---|---|---|
| Duffecy, 2019 | Antenatal | PHQ-9 | -1.075 | -2.383 | 0.233 | 7 | 4 | | 7.71 |
| Fonseca, 2020 | Postpartum | EPDS | -0.397 | -0.787 | -0.007 | 98 | 96 | | 11.97 |
| Forsell, 2017 | Antenatal | EPDS | -1.292 | -1.984 | -0.600 | 21 | 18 | | 10.73 |
| Jannati, 2020 | Postpartum | EPDS | -3.833 | -4.595 | -3.070 | 38 | 37 | | 10.39 |
| Loughnan, 2019 | Antenatal | PHQ-9 | -0.304 | -0.825 | 0.217 | 23 | 36 | | 11.49 |
| Loughnan, 2019 | Postpartum | PHQ-9 | -0.998 | -1.416 | -0.580 | 50 | 47 | | 11.88 |
| Milgrom, 2016 | Postpartum | BDI-II | -0.844 | -1.468 | -0.220 | 21 | 22 | | 11.04 |
| O'Mahen, 2013 | Postpartum | EPDS | -0.548 | -0.764 | -0.332 | 181 | 162 | | 12.44 |
| Van Lieshout, 2021 | Postpartum | EPDS | -1.870 | -2.119 | -1.620 | 165 | 192 | | 12.37 |
| | | | -1.213 | -1.794 | -0.632 | 604 | 614 | | |

**Figure 3.** Forest plot showing effect sizes for interventions delivered online ($n = 9$).

| Study name | Subgroup within study | Outcome | Std diff in means | Lower limit | Upper limit | Intervention | Control | Std diff in means and 95% CI | Relative weight |
|---|---|---|---|---|---|---|---|---|---|
| Ammerman, 2013 | Antenatal | EPDS | -0.889 | -1.315 | -0.463 | 47 | 46 | | 4.57 |
| Dimidjian, 2016 | Antenatal | EPDS | -0.240 | -0.775 | 0.294 | 24 | 31 | | 4.01 |
| Dimidjian, 2017 | Antenatal | PHQ-9 | -0.094 | -0.437 | 0.249 | 67 | 64 | | 5.01 |
| Hayden, 2012 | Antenatal | BDI | -0.422 | -1.112 | 0.269 | 20 | 14 | | 3.27 |
| Hou, 2014 | Postpartum | EPDS | -0.714 | -0.991 | -0.437 | 104 | 109 | | 5.33 |
| McKee, 2006 | Both | BDI-II | -0.075 | -0.687 | 0.538 | 21 | 20 | | 3.62 |
| Morrell, 2009 | Postpartum | EPDS>=12 | -0.353 | -0.626 | -0.081 | 231 | 231 | | 5.36 |
| Nasiri, 2018 | Postpartum | BDI | -2.029 | -2.686 | -1.372 | 26 | 28 | | 3.41 |
| Nejad, 2021 | Antenatal | DASS-21 | -1.685 | -2.274 | -1.096 | 30 | 30 | | 3.73 |
| Ngai , 2015 | Postpartum | EPDS >= 10 | -0.527 | -0.752 | -0.301 | 197 | 200 | | 5.56 |
| O'Mahen, 2013 | Both | BDI-II | -0.607 | -1.225 | 0.012 | 21 | 21 | | 3.59 |
| Prendergast, 2001 | Both | EPDS | 0.322 | -0.329 | 0.973 | 17 | 20 | | 3.44 |
| Rahman, 2008 | Both | HDRS | -0.391 | -0.529 | -0.252 | 418 | 400 | | 5.87 |
| Sikander, 2019 | Both | PHQ-9 | -0.300 | -0.484 | -0.116 | 223 | 211 | | 5.72 |
| Silverstein, 2011 | Antenatal | QIDS | -0.503 | -1.170 | 0.165 | 25 | 25 | | 3.37 |
| Burns, 2013 | Antenatal | EPDS | -0.966 | -1.739 | -0.193 | 16 | 13 | | 2.92 |
| Tandon, 2018 | Both | BDI-II | -0.213 | -0.714 | 0.288 | 40 | 25 | | 4.18 |
| Trevillion, 2016 | Both | EPDS | -0.640 | -1.303 | 0.023 | 24 | 26 | | 3.39 |
| Van Horne, 2021 | Postnatal | EPDS | -0.303 | -0.732 | 0.127 | 58 | 33 | | 4.56 |
| Yazdanimehr, 2016 | Antenatal | EPDS | -2.118 | -2.750 | -1.486 | 30 | 30 | | 3.53 |
| Chabrol, 2002, Prev | Postpartum | EPDS | -0.426 | -0.700 | -0.153 | 97 | 114 | | 5.35 |
| Chabrol, 2002, Treat | Postpartum | HDRS | -2.562 | -3.340 | -1.785 | 18 | 30 | | 2.91 |
| Cho, 2008 | Antenatal | BDI | -0.134 | -0.974 | 0.707 | 12 | 10 | | 2.67 |
| Cooper, 2003 | Postpartum | EPDS | -0.438 | -0.853 | -0.022 | 42 | 50 | | 4.63 |
| | | | -0.626 | -0.810 | -0.443 | 1808 | 1781 | | |

**Figure 4.** Forest plot showing effect sizes for interventions delivered to individuals ($n = 24$).

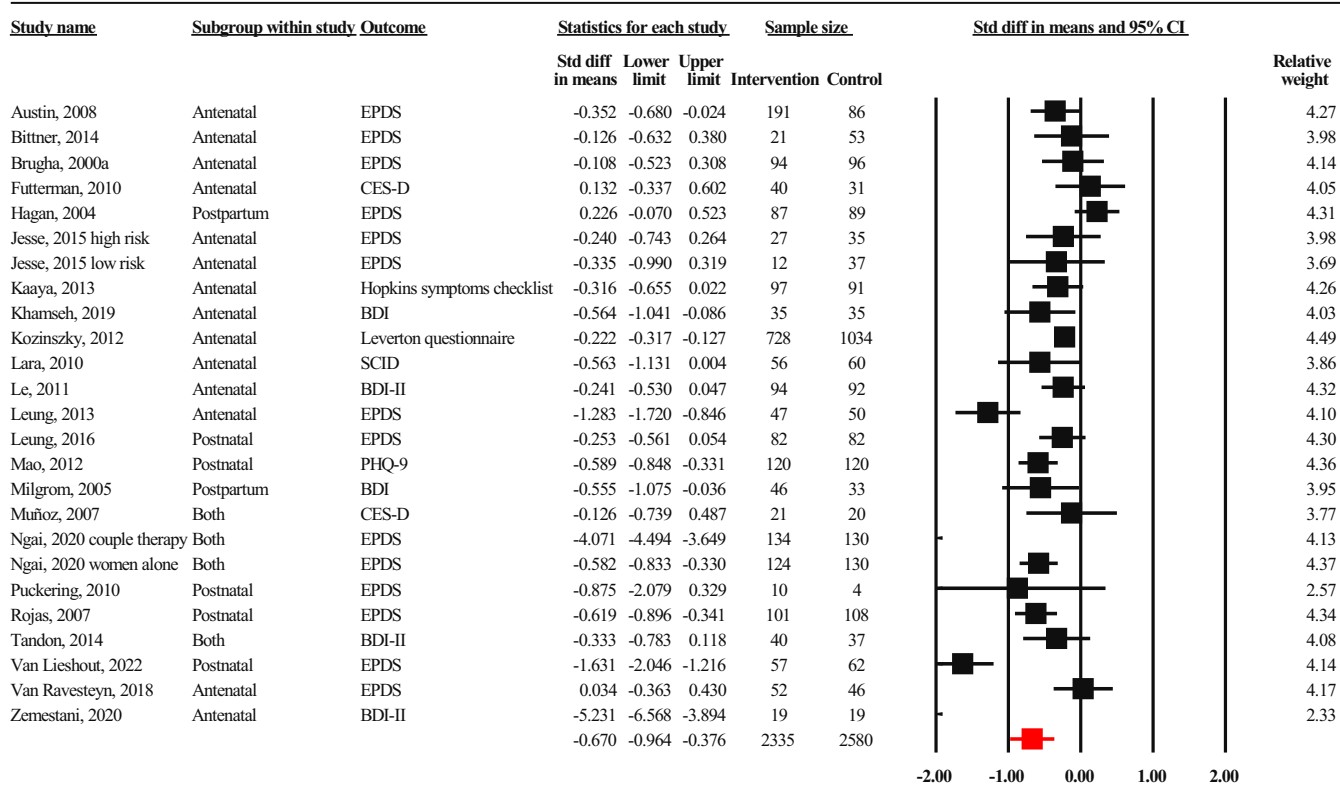

| Study name | Subgroup within study | Outcome | Statistics for each study | | | Sample size | | Std diff in means and 95% CI | | |
|---|---|---|---|---|---|---|---|---|---|---|
| | | | Std diff in means | Lower limit | Upper limit | Intervention | Control | | | Relative weight |
| Austin, 2008 | Antenatal | EPDS | -0.352 | -0.680 | -0.024 | 191 | 86 | | | 4.27 |
| Bittner, 2014 | Antenatal | EPDS | -0.126 | -0.632 | 0.380 | 21 | 53 | | | 3.98 |
| Brugha, 2000a | Antenatal | EPDS | -0.108 | -0.523 | 0.308 | 94 | 96 | | | 4.14 |
| Futterman, 2010 | Antenatal | CES-D | 0.132 | -0.337 | 0.602 | 40 | 31 | | | 4.05 |
| Hagan, 2004 | Postpartum | EPDS | 0.226 | -0.070 | 0.523 | 87 | 89 | | | 4.31 |
| Jesse, 2015 high risk | Antenatal | EPDS | -0.240 | -0.743 | 0.264 | 27 | 35 | | | 3.98 |
| Jesse, 2015 low risk | Antenatal | EPDS | -0.335 | -0.990 | 0.319 | 12 | 37 | | | 3.69 |
| Kaaya, 2013 | Antenatal | Hopkins symptoms checklist | -0.316 | -0.655 | 0.022 | 97 | 91 | | | 4.26 |
| Khamseh, 2019 | Antenatal | BDI | -0.564 | -1.041 | -0.086 | 35 | 35 | | | 4.03 |
| Kozinszky, 2012 | Antenatal | Leverton questionnaire | -0.222 | -0.317 | -0.127 | 728 | 1034 | | | 4.49 |
| Lara, 2010 | Antenatal | SCID | -0.563 | -1.131 | 0.004 | 56 | 60 | | | 3.86 |
| Le, 2011 | Antenatal | BDI-II | -0.241 | -0.530 | 0.047 | 94 | 92 | | | 4.32 |
| Leung, 2013 | Antenatal | EPDS | -1.283 | -1.720 | -0.846 | 47 | 50 | | | 4.10 |
| Leung, 2016 | Postnatal | EPDS | -0.253 | -0.561 | 0.054 | 82 | 82 | | | 4.30 |
| Mao, 2012 | Postnatal | PHQ-9 | -0.589 | -0.848 | -0.331 | 120 | 120 | | | 4.36 |
| Milgrom, 2005 | Postpartum | BDI | -0.555 | -1.075 | -0.036 | 46 | 33 | | | 3.95 |
| Muñoz, 2007 | Both | CES-D | -0.126 | -0.739 | 0.487 | 21 | 20 | | | 3.77 |
| Ngai, 2020 couple therapy | Both | EPDS | -4.071 | -4.494 | -3.649 | 134 | 130 | | | 4.13 |
| Ngai, 2020 women alone | Both | EPDS | -0.582 | -0.833 | -0.330 | 124 | 130 | | | 4.37 |
| Puckering, 2010 | Postnatal | EPDS | -0.875 | -2.079 | 0.329 | 10 | 4 | | | 2.57 |
| Rojas, 2007 | Postnatal | EPDS | -0.619 | -0.896 | -0.341 | 101 | 108 | | | 4.34 |
| Tandon, 2014 | Both | BDI-II | -0.333 | -0.783 | 0.118 | 40 | 37 | | | 4.08 |
| Van Lieshout, 2022 | Postnatal | EPDS | -1.631 | -2.046 | -1.216 | 57 | 62 | | | 4.14 |
| Van Ravesteyn, 2018 | Antenatal | EPDS | 0.034 | -0.363 | 0.430 | 52 | 46 | | | 4.17 |
| Zemestani, 2020 | Antenatal | BDI-II | -5.231 | -6.568 | -3.894 | 19 | 19 | | | 2.33 |
| | | | -0.670 | -0.964 | -0.376 | 2335 | 2580 | | | |

**Figure 5.** Forest plot showing effect sizes for interventions delivered to groups (*n* = 25).

### Moderator analyses: Intervention level characteristics

Moderator analyses for intervention-level characteristics yielded several important insights (Supplementary Table 8). Interventions for treatment (SMD = −0.94, 95% CI: −1.15 to −0.73) of perinatal depressive symptoms yielded significantly higher effect sizes than preventive ones (SMD = .−0.36, 95% CI: −0.65 to −0.07). Interventions offered as tested as stand-alone programs (SMD = −1.01, 95% CI: −1.24 to −0.79) performed better than those integrated into healthcare settings (SMD = −0.38, 95% CI: −0.63 to −0.14).

Effect sizes did not differ according to the delivery format, where no differences were observed between interventions delivered either through electronic means, face-to-face in groups, or individually (*Q* = 4.76 and *p* = 0.09). Delivery agents with varying disciplinary backgrounds, multidisciplinary teams, nonspecialists, online interventions, and those delivered by specialists, were effective. Although interventions delivered electronically and through specialists had slightly higher effect sizes, this did not reach statistical significance (*Q* = 4.05 and *p* = 0.26).

### Moderator analysis: Participant-level characteristics

Higher effect sizes were associated with interventions recruiting perinatal women with higher age (*b* = −0.07, SE = 0.01, and *p* = <0.001; Supplementary Table 9). While interventions with a higher proportion of perinatal women belonging to minorities, low-income levels, reporting poorer education, and recurrent episodes of depression yielded smaller effect sizes. The proportion of married or primiparous women in trials was not associated with effect sizes yielded by included interventions. Interventions delivered

during postnatal had a higher effect size than those delivered during the antenatal period, or during both periods; however, this was statistically non-significant.

### Moderator analysis: Active ingredients

When considering the theoretical underpinnings of included interventions, the dose of intervention was inversely associated with effect sizes (*b* = 0.016 and *p* < 0.01). Most of the trial evidence included in this review tested CBT interventions (*n* = 42), which yielded strong effect sizes (SMD = −0.70, 95% CI: −0.91 to −0.49). PST was tested in five trials and yielded comparable effect sizes (−0.71, 95% CI: −1.32 to −0.11). BA yielded moderate-strength effect sizes (SMD = −0.32, 95% CI: −1.05 to 0.42). However, evidence about these was inconclusive due to overlapping effect sizes, despite reaching statistical significance.

Among intervention ingredients, using more behavioral ingredients in CBT interventions yielded high effect sizes (*b* = −0.079 and *p* < 0.01). An inverse trend was noted for interventions including reinforcement-related ingredients (*b* = 0.2 and *p* < 0.01). Interventions including a higher number of cognitive and interpersonal ingredients, parenting skills, psychoeducation, exercise, in-session techniques, nutrition, and substance use-related counseling did not yield statistical significance (Supplementary Tables 4−9). When individual active ingredients were considered, the presence or absence of interpersonal, cognitive, and behavioral ingredients did not alter effect sizes. Interventions utilizing *identifying affect* and *self-awareness* strategies yielded larger effect sizes than their counterparts (Supplementary Tables 4−10).

## Discussion

The present systematic review presents up-to-date evidence regarding the effectiveness of CBT for PND. It delineates several interesting insights for optimizing CBT-based interventions for PND. We found that CBT interventions, including third-wave cognitive therapies, are highly effective in preventing and treating PND. CBT can be delivered effectively to individuals and in groups or online web or app-based software. The delivery of CBT can also be tailored according to the resources available, for instance, by employing specialists or nonspecialists' delivery agents. Interventions integrated into healthcare settings and utilizing the available infrastructure may be less effective than stand-alone programs. Perinatal women experiencing adverse events and health inequalities report smaller effect sizes when treated with CBT. The effectiveness of CBT also depends on several intervention-level characteristics.

CBT interventions yielded strong effect sizes for treatment and moderate strength effect sizes for preventing PND. These findings are corroborated by previous meta-analyses, which have yielded similar effect sizes for CBT interventions for PND (Rahman et al., 2013; Sockol, 2015; Rahman et al., 2018; Waqas et al., 2022b). CBT interventions are also recommended by the US Preventive Services Taskforce and the WHO (Rahman et al., 2018; Curry et al., 2019). Previous evidence has shown that CBT-based interventions are effective for PND and generally acceptable to stakeholders, delivery agents, and end-consumers (Morrell et al., 2009; Rahman et al., 2018). CBT interventions can be tailored to settings depending on the availability of resources. Both the National Institute for Health and Care Excellence (NICE) and the WHO recommend a stepped-care approach to treating PND (Rahman et al., 2018; National Institute for Health and Care Excellence, 2020; Delgadillo et al., 2022), ranging from self-help psychoeducational materials to low-intensity and high-intensity psychotherapies.

There has been an increasing focus on preventing PND. Recently, based on evidence from high-income countries, The U.S. Preventive Services Task Force (USPSTF) has recommended the use of CBT and counseling interventions for PND (Curry et al., 2019). Although the WHO have recommended that all perinatal women should be offered psychosocial interventions to develop coping, stress management, and social skills (Guidelines Review Committee, 2022), women at high risk of developing PND should be offered psychological interventions such as CBT and interpersonal therapy. The provision of these interventions should be allowed as per the availability of resources and women's preference. Our systematic review corroborates this evidence and presents CB-based approaches (both specialist and nonspecialist delivered) as effective in preventing depression during the perinatal period. We also found that CB-based approaches yield good effect sizes across all modes of delivery (electronic, individual, or group). This flexibility in delivery increases the utility of CB-based approaches in different settings.

While reviewing the intervention level characteristics, several valuable insights were revealed. First, these interventions work when delivered antenatally or postnatally, with little difference in effect sizes. This finding does not agree with our previous systematic review of preventive interventions where a higher effect size was demonstrated for interventions starting early during the antenatal period (Waqas et al., 2022b). This finding also contradicts Sockol's meta-analysis of 26 treatment interventions, where more considerable reductions were noted for interventions initiated during the postpartum period or across the perinatal period (Sockol, 2015).

Second, CBT interventions delivered either to individuals or groups or online yield similar strength of effect sizes, also corroborated by previous systematic reviews (Sockol, 2015).

Interventions integrated into healthcare settings and utilizing the available infrastructure may be less effective than stand-alone programs. This interesting insight emphasizes the importance of effective implementation measures to ensure adequate implementation, supervision, and competency measures (Zafar et al., 2016; Ahmad et al., 2020). A critical case study in this context is that of the Thinking Healthy Programme developed by one of the co-authors (Rahman et al., 2008). It is a highly effective low-intensity CBT-based intervention that has been endorsed by the WHO for the treatment of PND (World Health Organization, 2015). Integrated into the primary healthcare system, it employed lady health workers as the delivery agents (Rahman et al., 2008). In the following years, a trial was run to test the effectiveness of Thinking Healthy Program (THP) delivered by peers with lived experience of PND (Sikander et al., 2019a). These innovations ensured that the THP remained cost-effective and acceptable to the stakeholders. In addition to innovations in delivery, newer approaches in enhanced supervision, competency assessments, and training at a large scale were also tested to ensure seamless implementation of the THP in communities (Zafar et al., 2016; Ahmad et al., 2020).

While reviewing the active ingredients of included interventions, several insights emerged. In comparison with the face-to-face delivered CBT programs, the ingredient of empathy was missing in electronically delivered interventions. However, this collection of interventions yielded pooled effect sizes comparable to the interventions delivered face to face. This is an important finding as empathy is the foundation for an effective therapeutic patient alliance (Morrell et al., 2009). Therefore, there is a need to open further the black box of the causal mechanisms at play that drive the effectiveness of electronic interventions without the opportunity to build an empathy-based therapeutic relationship. Another interesting finding was that longer interventions were associated with a decrease in effect size. This association may be driven by burnout among either the patient or therapist. We also investigated the dosage density of therapeutic strategies and their association with effect sizes. Only one significant association emerged, where an increase in behavioral ingredients in a therapeutic program led to an increase in effect size. This strengthens the previous notion that (Kahl et al., 2012) efficacy of the cognitive therapy depends critically on the behavioral activation component rather than its content-oriented cognitive approaches. However, this is inconclusive and warrants further investigation, especially for PND, due to the lack of RCTs, for instance, those comparing efficacy of BA with classical CBTs. This is indeed an important area for further research.

Lastly, we found that younger perinatal women reported poor education and belonging to lower economic, and minority ethnic classes reported a lower reduction in PND symptoms. This finding is significant and highlights the importance of contextual factors affecting community health and community-oriented policies and initiatives. Multidisciplinary approaches, such as mass education and poverty alleviation initiatives, are required to tackle this issue. In this regard, Banerjee and colleagues' Nobel prize-winning multi-faceted program rooted in developmental economics is a crucial case study (Banerjee et al., 2015). Such initiatives are necessary to curb the effects of societal adversities impeding the efficacy of psychological treatments. This has been shown in a huge body of literature demonstrating the complexity of PND among women facing adversities (Ashman et al., 2008; Bao et al., 2016; Chae et al., 2020).

Meta-regression analyses revealed a weakly inverse association between the proportion of women with a history of mental health problems and intervention effect size. There is unequivocal evidence that complex presentations of PND (increased severity, relapsing, and recurrent) are associated with poorer treatment response (Ahmed Waqas and Rahman, 2023). A recent systematic review of observational studies demonstrates that perinatal women with complex and more severe forms of PND report more psychosocial adversities (Ahmed Waqas and Rahman, 2022). Moreover, if left untreated, such PND symptoms contribute to intergenerational transfer of inequities; whereby children born to women with complex PND report poorer academic, mental, and physical health outcomes. Despite a plethora of observational research evidence, investigators have not yet focused on the development of bespoke interventions for either preventing relapse or treating recurrent episodes of perinatal depressive disorder. This is also true for pharmacological trials where little evidence is present for the prevention of relapse of depression during the perinatal period (Molyneaux et al., 2018). Evidence is emerging, however, where a recent two-arm, parallel-design RCT tested a parenting video-feedback therapy intervention added to CBT in the treatment of persistent postpartum depression (Stein et al., 2018). The NICE recommends high-intensity psychotherapies or antidepressants for women at a high risk of relapse (National Institute for Health and Care Excellence, 2020).

## Strengths and limitations

This systematic review has several strengths. First, this systematic review and meta-regression analysis provide a comprehensive and up-to-date estimate of the effectiveness of CBT. It provides reliable estimates of the effectiveness of CBT delivered by specialist and nonspecialist workforces. Furthermore, this review utilizes a large pool of RCTs. This allowed us to investigate the moderating effects of intervention and patient-level characteristics in detail. We also present novel findings on the active ingredients of CB-based approaches by leveraging the distillation and matching framework. Effects of dose density and active ingredients comprising CB interventions yielded valuable insights.

However, despite its strengths, this review has several limitations. First, conducting distillation and matching framework exercises to map active ingredients of therapies is complex. The accuracy of this endeavor depends on the information regarding the content of interventions provided in primary studies. Interventions such as the THP (developed by the co-author AR) (Rahman et al., 2008) provided details and content of the intervention in open-access manuals (World Health Organization, 2015). This approach is important and aids in future evidence synthesis studies and reproducibility and adaptability in different cultures. These analyses are also limited by the observational nature of meta-regression analyses used to study moderators of CB interventions. Therefore, this evidence should be interpreted with caution.

The present meta-analysis utilized subgroup analyses to compare the effectiveness of CB-based approaches utilizing specific active ingredients. These analyses can be improved by using meta-analytic structural equation modeling approaches (Harrer et al., 2021). The use of these complex methods can aid in our understanding of causal mediation mechanisms in psychotherapies.

Another key limitation inherent to using meta-regression analyses is the use of across-trial data and aggregated values for the participant and intervention-level characteristics for analyses. Such analyses are limited due to inherent aggregation bias and may not

reflect actual treatment-covariate interactions (Kelley and Kelley, 2012; Huh et al., 2019). These limitations, in theory, can be offset by using two-stage Individual Participant Data Meta-analysis (IPDMA) approaches that use within-trial information to estimate treatment–covariation interactions (Kelley and Kelley, 2012). IPD-MAs involving a large pool of datasets are time and resource intensive; however, we encourage researchers to utilize these approaches in the future. A recent example of this approach is Furukawa and colleagues' work (Furukawa et al., 2021), which presents a web application to estimate relative efficacies, and additive and synergistic effects yielded through combinations of specific and nonspecific components in Internet-delivered CBT interventions in the context of patient-level variables. Future meta-investigations should also consider utilizing realist evaluation using both quantitative and qualitative approaches to distill important insights on CBT for PND.

Furthermore, the subgroup and meta-regression analyses in this systematic review were run for a limited number of participant-level and intervention-level factors. Many other factors such as experience of intimate partner violence (Keynejad et al., 2020), family structure, social support networks, and chronicity of PND are important moderators and should be considered in future reviews (Waqas et al., 2022a). Moreover, researchers should consider collecting detailed data on moderators of treatment for PND in their future trials. We focused on CB-based approaches to meta-analyze a homogeneous set of interventions in the present systematic review. Other psychotherapeutic modalities should be reviewed in future meta-analyses, keeping in mind the clinical and statistical heterogeneity often encountered in psychotherapy literature.

## Conclusion

CBT are highly effective in reducing the severity of PND. Most of the trial evidence included in this review tested classical CBT approaches. And there is limited evidence for third-wave CBT for PND. CBT is effective when delivered across individual, group, and electronic platforms and thus can be tailored according to the financial and human resources available. Longer duration CBT interventions may not necessarily be more effective than shorter ones. Furthermore, CBT-based interventions should consider including various behavioral ingredients to maximize intervention benefits.

**Open peer review.** To view the open peer review materials for this article, please visit http://doi.org/10.1017/gmh.2023.8.

**Supplementary materials.** To view supplementary material for this article, please visit http://doi.org/10.1017/gmh.2023.8.

**Data availability statement.** All data associated with this manuscript are available as supplementary files.

**Author contributions.** This systematic review and meta-analysis was conceived by A.W. and A.R. A.W. and A.R. wrote the protocol and registered it in PROSPERO. A.W. & P.A. searched the databases and performed screening of titles and abstracts and full texts for eligibility. A.W. and S.N. extracted data on the characteristics of intervention and population. A.W., P.A., and S.W.Z. extracted data pertaining to components of interventions. A.W. and S.N. extracted quantitative data. A.W. conducted the meta-analysis. A.W. wrote the initial draft of the manuscript. All authors critically reviewed the manuscript and approved it for submission.

**Financial support.** This study has not received any funding.

**Competing interest.** The authors have no conflict of interest to report.

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
