## [Reviewer Report]

*Comments to Author*: Overall this is a very rigorous robust review which could be further strengthened with more articulations of the authors’ views on how to tailor CBT programmes for women who do not respond and other minor suggestions below.

The background is pertinent up-to-date and thoughtful. There are four aims of this review – three focused on effectiveness and active ingredients of CBT and the contexts for interventions – it would be helpful to clarify what was meant by contexts here in the background. The third aim was to explore factors driving prognosis and prescriptive factors among women undergoing CPD and, again, it would be helpful to clarify some of the terminology here in the background such as prescriptive factors.

The methods have many strengths including pre-registration of the protocol. This is a very extensive review as it includes both prevention and treatment of perinatal depression, though only the primary outcomes are reported here.

The results were written clearly and were well presented. I note that there were several important patient level characteristics included but some of the more sensitive variables such as intimate partner violence or childhood maltreatment were not and I would be surprised if none of the studies included such variables. I therefore wondered if it were possible to include more detail and include these and/or to mention this limitation in the discussion.

The Thinking Healthy Programme was often mentioned (not surprisingly in view of the impact it has had), with no explicit mention that this is the intervention developed and implemented by this review’s authors. My view is that it would be better to be very explicit therefore when mentioning the programme in both the results and discussion e.g. “their intervention” could be referred to as “our intervention” for transparency.

Some women e.g. those with a history of depression were found to be less likely to respond and it would be helpful if these authors, who are very experienced in delivering this intervention in low-income settings, could articulate their thought on how to help such women who continue to experience depressive symptoms. The discussion could also be strengthened by some thoughts on why this review’s findings differ from other systematic reviews.

Finally, in terms of what works best for whom, there are other methodological approaches which may be helpful in thinking about which individual woman would respond best to what treatment. For example, realist evaluations use both qualitative and quantitative methods and the authors could also discuss other methodological approaches.

---

## [Reviewer Report]

*Comments to Author*: This is a thoughtful and thorough review of CBT-based psychotherapies for treatment and prevention of perinatal depression. The authors state that while there has been sufficient research to demonstrate the efficacy of CBT in treatment perinatal depression, there is a gap in summarizing the research on prevention of perinatal depression in particular and in ‘unpacking’ the ‘mechanisms of action to understand what approaches work best for whom.’

A few areas where the authors could provide additional clarification and information include:

- Clarifying why a specific focus on CBT-based interventions and what is lost by not including non-CBT-based psychotherapies and can this be mentioned in the limitations.

- It would be helpful in the results to more clearly specify and discussion intervention vs. prevention trials – particularly given the bigger gap in the prevention literature.

- The summary of the moderator results – which focus on meeting the goal of unpacking mechanisms to support what works best for whom - seem somewhat basic and could benefit from a deeper dive, particularly to support different programs using these results to inform their services. For example, providing more details about variation by who/how the interventions were delivered (there seems some variation despite all being effective) and whether other characteristics of the participant were explored in the studies (including history of trauma and violence, comorbid mental and behavioral health conditions).

---

## [Reviewer Report]

*Comments to Author*: The reviewers commend your systematic review but each have recommendations to strengthen the paper. Please address all the reviewers' comments.